

# Operational numerical weather prediction with ICON on GPUs (version 2024.10)

Xavier Lapillonne[1], Daniel Hupp[1], Fabian Gessler[2], André Walser[1,†], Andreas Pauling[1], Annika Lauber[2], Benjamin Cumming[5], Carlos Osuna[1], Christoph Müller[1], Claire Merker[1], Daniel Leuenberger[1], David Leutwyler[1], Dmitry Alexeev[3], Gabriel Vollenweider[1], Guillaume Van Parys[1], Jonas Jucker[2], Lukas Jansing[1], Marco Arpagaus[1], Marco Induni[5], Marek Jacob[4], Matthias Kraushaar[5], Michael Jähn[2], Mikael Stellio[2], Oliver Fuhrer[1], Petra Baumann[1], Philippe Steiner[1], Pirmin Kaufmann[1], Remo Dietlicher[2], Ralf Müller[6], Sergey Kosukhin[7], Thomas C. Schulthess[5], Ulrich Schättler[4], Victoria Cherkas[2], and William Sawyer[5]

[1]Federal Office of Meteorology and Climatology MeteoSwiss, Switzerland
[2]Center for Climate Systems Modeling C2SM, ETH Zurich, Switzerland
[3]NVIDIA, Zürich, Switzerland
[4]Deutscher Wetterdienst, Offenbach am Main, Germany
[5]Swiss National Supercomputing Centre, ETH Zurich, Switzerland
[6]Deutsches Klimarechenzentrum, Hamburg, Germany
[7]Max Planck Institute for Meteorology, Hamburg, Germany
[†]deceased, 19 February 2025

**Correspondence:** Xavier Lapillonne (xavier.lapillonne@meteoswiss.ch)

**Abstract.** Numerical weather prediction and climate models require continuous adaptation to take advantage of advances in high-performance computing hardware. This paper presents the port of the ICON model to GPUs using OpenACC compiler directives for numerical weather prediction applications. In the context of an end-to-end operational forecast application, we adopted a full-port strategy: the entire workflow, from physical parameterizations to data assimilation, was analyzed and ported

5 to GPUs as needed. Performance tuning and mixed-precision optimization yield a 5.6x speed-up compared to the CPU baseline in a socket-to-socket comparison. The ported ICON model meets strict requirements for time-to-solution and meteorological quality, in order for MeteoSwiss to be the first national weather service to run ICON operationally on GPUs with its ICON-CH1-EPS and ICON-CH2-EPS ensemble forecasting systems. We discuss key performance strategies, operational challenges, and the broader implications of transitioning community models to GPU-based platforms.

## 1 Introduction

Numerical weather prediction (NWP) plays a critical role in our society, supporting applications ranging from renewable energy management to the protection of life and property during severe weather events. The growing demand for accurate weather forecasts requires the use of advanced computational techniques to improve the performance of NWP models. With greater computing power, models can run at higher resolutions, capturing finer-scale atmospheric processes and local weather phenom-

15 ena that coarser models cannot resolve (Palmer, 2014; Bauer et al., 2015). This could improve the accuracy of predictions for





severe weather events such as thunderstorms, hurricanes, and heavy rainfall. Furthermore, greater computational power allows for more frequent updates and the execution of larger ensemble forecasts, thereby enhancing forecast reliability through quantification of uncertainty. In recent years, graphics processing units (GPUs) have emerged as a cornerstone of high-performance computing (HPC), offering high throughput and improved energy efficiency through massive parallelism. However, in order to benefit from the latest advancements in HPC technology, weather and climate models must be adapted to operate effectively on this hardware. Furthermore in the context of emerging machine learning for weather forecasting having a GPU capable model is of great interest since both the Machine Learning algorithms as well as the traditional NWP model can run on the same GPU hardware allowing for synergies in terms of system investment.

Weather and climate models are typically large community codes comprising millions of lines of Fortran or C/C++ code. Adapting such a large code base to GPUs requires significant effort. While many attempts have been made to port individual components, only a handful of these models have been fully ported and are being used in production on GPU-based or hybrid systems. Different groups have adopted various strategies. Some early work includes the Japanese ASUCA model (Shimokawabe et al., 2011) which has been ported with a CUDA (Nickolls et al., 2008) rewrite. Another approach is to use compiler directives such as OpenACC or OpenMP for accelerators. The advantage of this approach is that it can be incrementally inserted into existing code and may be more easily accepted by the modeling community. The COSMO (Consortium for Small-scale Modeling) model was ported using a combination of a domain-specific language rewrite and OpenACC directives, achieving substantial speed-ups on GPU systems with respect to the CPU baseline (Lapillonne and Fuhrer, 2014; Fuhrer et al., 2014). It was also the first model used by a national weather service for operational numerical weather prediction on GPUs. More recently, the MesoNH (non-hydrostatic mesoscale atmospheric) model was ported to GPUs using compiler directives (Escobar et al., 2024), while the Energy Exascale Earth System (E3SM) was entirely rewritten using C++ and the Kokkos library (Donahue et al., 2024). These diverse efforts underscore the growing necessity of exploiting modern hardware architectures, such as GPUs, to meet the increasing computational demands of weather and climate modeling.

To also take advantage of advances in hardware architectures, the ICON (Icosahedral Nonhydrostatic) model (Zängl et al., 2015) – a state-of-the-art model for weather and climate simulations – was successfully ported to GPU architectures using OpenACC compiler directives applied to the existing Fortran-based code. The initial effort targeted climate applications (Giorgetta et al., 2022), leaving components essential for NWP unported. Building on this foundation, we extend the GPU support to weather applications by porting the missing components using OpenACC directives. Although several other configurations are ported and supported on GPU, this work particularly focuses on the operational setup at the Swiss National Weather Service MeteoSwiss. Our comprehensive approach spans the entire operational workflow, from initialization to output, while meeting stringent time-to-solution requirements of various products such as forecasts for aviation or other NWP-specific applications.

## 2 The ICON model

The ICON (Icosahedral Nonhydrostatic) model, by (Zängl et al., 2015), is a climate and numerical weather prediction system developed through a partnership involving the German Weather Service (DWD), the Max Planck Institute for Meteorology





(MPI-M), the German Climate Computing Center (DKRZ), the Karlsruhe Institute of Technology (KIT), and the Center for
Climate Systems Modeling (C2SM). Besides climate and research applications, ICON is used operationally by the DWD,
MeteoSwiss and several national weather services of the COSMO consortium for numerical weather prediction. The model
employs an icosahedral grid structure, which divides the globe into triangular cells, providing quasi-uniform resolution and
eliminating the pole problem inherent to traditional latitude-longitude grids. This grid structure supports local refinement, en-
abling higher resolution in areas of interest without compromising global coverage. The model uses a finite-volume discretiza-
tion method on these triangular cells, ensuring mass conservation and accurate representation of atmospheric dynamics. The
coupling between dynamics and physics in ICON follows a split-explicit strategy, where the fast dynamical core is sub-stepped
relative to the physics time step, allowing for stable integration while maintaining computational efficiency.

For parallelization, ICON adopts a horizontal domain decomposition strategy, where the computational domain is divided
into smaller subdomains distributed across multiple processors. This approach, combined with advanced data structures and
efficient communication protocols, ensures scalability and optimal performance on massively parallel super-computing archi-
tectures. The vertical discretization in ICON is based on a terrain-following hybrid coordinate system which combines the
advantages of pressure-based and height-based coordinates. This system allows for a more accurate representation of the atmo-
spheric boundary layer and better resolution of vertical atmospheric processes, particularly in regions with complex topography
such as the Alps.

## 65 3 GPU port for NWP

### 3.1 Overview

The GPU port of the ICON model is based on OpenACC compiler directives, which are added as comments to the original
Fortran source code; see Listing 1. The choice of a directive based approach was decided over a re-write in a GPU specific
language like CUDA or a DSL, mainly because of it's broader acceptance by the ICON community, increasing the chance
to re-integrate the work in the main code. Further more the OpenACC directives were chosen over OpenMP for accelerator,
because when starting this porting effort and evaluating available technologies the compilers supporting OpenMP were less
mature. The OpenACC directives guide the compiler in generating GPU code, reducing the need for manual intervention. One
of the key aspects of the port is to reduce data movement between the host CPU and the GPU memory while also maximizing
the utilization of the GPU's computational resources.

**Listing 1.** Example of an OpenACC block from the microphysics component, with parallelization along `nproma`, `jc` loop, and `nlev`, `jk`
loop

```
1:      !$ACC PARALLEL DEFAULT(PRESENT) ASYNC(1)
2:      !$ACC LOOP GANG VECTOR COLLAPSE(2)
3:      DO jk=1,nlev
4:          DO jc=i_startidx,i_endidx
5:              prm_diag%tt_lheat(jc,jk,jb) = prm_diag%tt_lheat(jc,jk,jb) - p_diag%temp(jc,jk,jb)
```




| 6: | ENDDO |
| 7: | ENDDO |
| 8: | !$ACC END PARALLEL |

In atmospheric models, the arithmetic intensity, i.e., the ratio of computation (floating-point operations) to memory accesses, is generally low (Adamidis et al., 2025). As a result, only porting isolated kernels to the GPU yields little benefit. Instead, most model components must be ported together, which we refer to as a full port strategy. Timing measurements confirm that transferring all the variables from the CPU to the GPU memory takes approximately as long as performing one full time step on the GPU. To achieve any performance gain using a GPU under these conditions, it is essential that such data transfers are

avoided at every time step. Instead, data transfers between CPU and GPU should be limited to infrequent operations, such as writing output to disk. Accordingly, the ICON GPU port is designed such that after the initialization phase on the CPU, all data are copied to the GPU and during the time loop all components called at high temporal frequency are executed on the GPU (Fig. 1). If any code needs to run on the CPU within the time loop, a data copy is added from GPU to CPU.

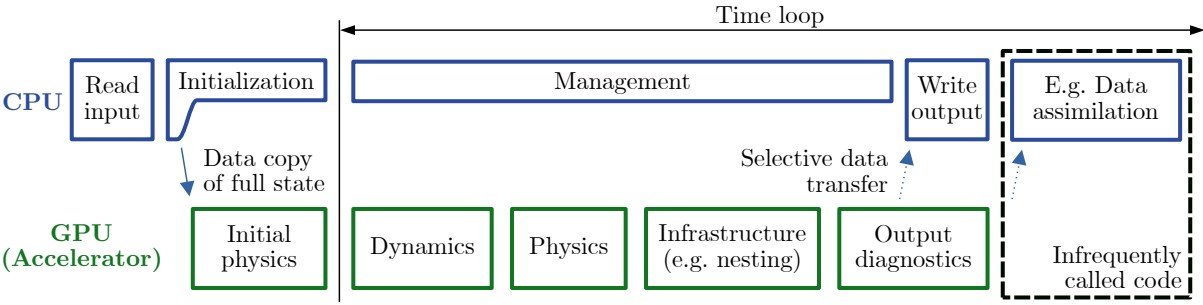

**Figure 1.** GPU port of the ICON model. After initialization on the GPU all the main components of the time loop are run on the GPU.

### 3.2 Strategy and performance consideration

ICON operates on a three-dimensional computational domain: in the horizontal grid cells are enumerated with a space-filling curve, which is split into `nblocks` blocks of user-defined block size `nproma`, while the vertical direction comprises `nlev` levels. Most arrays in ICON follow the index ordering `(nproma, nlev, nblocks)`, possibly with additional dimensions of limited size. In the GPU port, `nproma` is set as large as possible, ideally such that all cell grid points of a computational domain, including first and second-level halo points, fit into a single block, yielding `nblocks=1`. This design choice reduces

the complexity of the porting process: parallelization is required only along the `nproma` and `nlev` dimensions, while `nblock` loop can be omitted, since its value is either one or very small. This particularly reduces the call tree size inside a parallel region, since many parameterizations are called inside the block loop. Loops over `nproma` are usually the innermost loops and rarely contain additional subroutine calls. Although `nblocks=1` should not be a requirement from the OpenACC standard point of view, in practice many compiler issues and limitations were circumvented by avoiding calling nested subroutines in accelerated





regions. The `nproma` dimension, with unit stride in memory, is the main direction of fine-grain parallelism and is associated with the OpenACC keyword `vector` to ensure coalesced memory access.

In our porting approach, all components in the time-stepping loop (with minor exceptions) run on the GPU, while components in the initialization run on the CPU (cf. Fig. 1). A challenge arises from the fact that both initialization and time-stepping invoke shared low-level routines. To distinguish between the execution context, an additional logical argument, `lacc`, is intro-

duced and used as a conditional guard on all parallel and data regions within the shared routines.

The GPU implementation assumes one compute MPI task per GPU. The original communication library is ported to GPU and supports GPU-to-GPU (G2G) communication, enabling efficient data exchange between GPUs during parallel execution. This comprehensive approach to porting ICON to GPUs aims to maximize performance gains and reduce to the minimum data transfer.

### 3.3 Basic Optimization

Multiple general optimization strategies are considered for the initial GPU port of ICON. The first design choice is to apply `ACC LOOP VECTOR` to the innermost loop to allow contiguous memory accesses. The inner `nproma`-loop can be collapsed with the outer `nlev` loop when appropriate, potentially improving performance. Collapsing loops allows both the inner and outer loops to be parallelized together, which is possible if there are no data dependencies across levels.

The second optimization is to merge multiple loops nested together into one large parallel region. This reduces the kernel launch overhead. However, a drawback of this approach is that it often prevents the use of the `COLLAPSE` clause and thus reduces the available parallelism, as every parallel loop in the whole region needs to have the same bounds, which is not always the case in the vertical direction. Nevertheless, for ICON, larger parallel regions generally yield better GPU performance. When dependencies exist between the different loop nests inside a parallel region, the `GANG(STATIC: 1)` clause is required. This

clause ensures that the same `GANG` indices will be executed consecutively in each sub-loop, which is otherwise not guaranteed. Since the `GANG(STATIC: 1)` clause does not degrade performance noticeably, the clause is always added when multiple loop nests are in a larger parallel region. This practice ensures robustness against future modifications by domain scientists who may introduce inter-loop dependencies.

An alternative to `COLLAPSE` is the use of the `TILE` directive. `TILE` splits the iteration space into blocks of user-defined size.

This can improve cache-usage, especially in cases for neighbor accesses or for memory transposes. For instance, the execution time for this parallel region can be reduced from $19\,\mathrm{ms}$ to $2.5\,\mathrm{ms}$ by using the `TILE` directive instead of `COLLAPSE(3)`, see Listing 2. The choice of tile sizes is hardware-specific and has been selected to yield optimal performance on NVIDIA GPUs, which is generally consistent across most NVIDIA architectures. In this case, it has also been demonstrated that even a suboptimal tile size can outperform a plain loop collapse.

**Listing 2.** Example for the use of the OpenACC tile directive

```
1: !$ACC PARALLEL DEFAULT(NONE) ASYNC(1)
2: !$ACC LOOP GANG VECTOR TILE(2, 64, 1)
3: DO JLEV = 1, KLEV
```





```
  4: !cdir unroll=4
  5:   DO JI = 1, JPGPT
  6:     DO JLON = KIDIA, KFDIA
  7:       POD(JI,JLEV,JLON) = ZTAU(JLON,JI,JLEV)
  8:     ENDDO
  9:   ENDDO
 10: ENDDO
 11: !$ACC END PARALLEL
```

Furthermore, advanced optimizations which are specific to different parts of the code are described in Sect. 6.2.

### 3.4 Dynamics

The Dynamics, or dynamical core of the model, solves the equations governing atmospheric flow. Most components of the dynamical core are shared between the climate and NWP configurations, so only a few additional components had to be ported in addition to the initial work described in Giorgetta et al. (2022). Most loops are parallelized along the `nproma` and `nlev` directions when there is no vertical dependency. The code has been GPU-optimized while preserving portability. Although OpenACC directives support execution on both CPU and GPU targets, around ten instances in the code use architecture-specific variants to achieve optimal performance on each platform. These conditional branches enable tailored implementations where the GPU and CPU exhibit substantially different performance characteristics.

### 3.5 Tracer Transport

The transport module is responsible for the large-scale redistribution of water substances, chemical constituents, or aerosols, for example pollen, by solving the tracer mass continuity equation. The Tracer advection is divided into independent horizontal and vertical advection using the Strang splitting approach see (Reinert, 2020). Both horizontal and vertical transport use semi-Lagrangian algorithms as described in (Reinert and Zängl, 2021). The computational structure of the transport routines resembles that of the dynamical core, involving access patterns from the current edge/cell to neighboring edges or cells. This similarity allows for the adoption of a comparable GPU porting strategy. Different transport algorithms with different computational cost/accuracy trade-offs can be chosen for different tracers. For example, cloud ice, cloud water, precipitation, graupel are transported using the same scheme, while water vapor is treated with a higher-order horizontal advection method. This high-order scheme poses the greatest challenge for GPU porting using OpenACC, as it involves indirect addressing using index lists. To address this, a tailored GPU implementation was developed to ensure efficient execution on GPUs.

### 3.6 Physics

The so-called physical parameterizations are additional components that describe physical processes not represented by the equations of the dynamics, such as sub-grid scale turbulence, radiation or the physical processes associated to cloud formation





and precipitation. These parameterizations are computed on the three-dimensional model grid and are invoked frequently, some at every time step, thus requiring GPU porting. Due to anisotropy in the atmosphere and the timescales of the physical processes relative to atmospheric flow, horizontal interactions can be neglected for most parameterizations. As a result, they can be formulated as column-independent computations. This makes them very attractive for parallelization, as the parameterizations

can be trivially parallelized along the horizontal direction, `nproma` in our case. Many parameterizations, however, do have vertical dependencies, e.g., when the vertical loop must be computed sequentially. This is true, for example, in the main computations of the microphysics and the turbulence. All parameterizations required for the main NWP applications have been ported to GPU.

### 3.6.1 Treatment of the soil tiles

In ICON, subgrid heterogeneity of the land surface is represented using a so-called tile approach. For example, a given grid point may consist of 50% forest and 50% grass, and the tile composition may change dynamically over the simulation. The ICON implementation is such that at each time step a list is created for each type of tile, which are then computed one after the other. Executing each tile type sequentially on the GPU would lead to poor performance, since some tile types may contain too few grid points on a given subdomain to fully utilize the GPU. To circumvent this issue, each tile type is run in an independent

queue, equivalent to a CUDA stream on NVIDIA GPUs, using the `ASYNC(stream)` construct. In combination with CUDA Graphs, an optimization further described in Sect. 6.2.5, this approach yields good performance for parameterization using tiles.

### 3.6.2 Radiation

The radiation scheme ecRad (Hogan and Bozzo, 2016), developed at the European Centre for Medium-Range Weather Fore-

casts (ECMWF), is operationally used in ECMWF's Integrated Forecasting System (IFS) and is also employed in ICON. The code structure and data layout of ecRad differ significantly from those in the rest of ICON. In addition, its memory consumption is substantial, as it processes approximately 300 wavelengths per grid point.

Thus, a solution is needed to manage memory consumption, and the porting strategy had to be adapted to accommodate the distinct data layout and code structure. To address the memory consumption, we introduced sub-blocking at the interface

between ecRad and ICON. The sub-blocking divides the grid points on the reduced grid into `nproma_sub`-sized batches, which are computed sequentially. This approach ensures that only one radiation sub-block must be held in memory at any given time (see Figure 2). While this method effectively controls memory usage, it reduces the available parallelism in ecRad. In practice, this trade-off between memory usage and parallelism must be carefully balanced to ensure the computation fits within the available GPU memory while still achieving high hardware utilization.

The data layout in ecRad differs from that used elsewhere in ICON (see Sect. 3.2). ecRad operates on a three-dimensional domain: the horizontal dimension is the same as for the other part of ICON but is further split into subblocks `nblcks_sub` of the size `nproma_sub`, see Fig. 2; the vertical levels from the other dimension of size `nlev` are the same as in ICON; additionally, a third dimension spans the wavelengths `ng`. The index order is (`ng`, `nlev`, `nproma_sub`). This layout





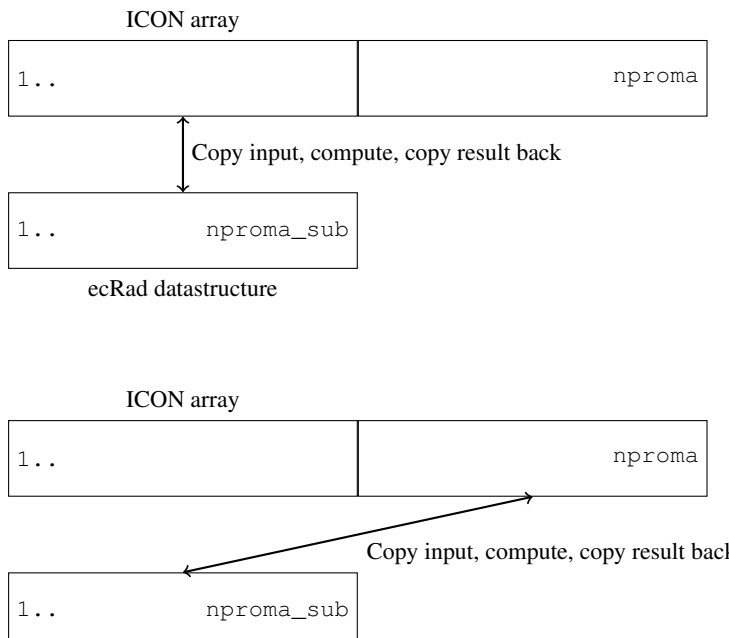

**Figure 2.** Example of two radiation subblocks that are used in the ecRad interface to compute each subblock sequentially.

presents a challenge in using the parallelism. The wavelength dimension offers limited parallelism – on the order of hundreds

of wavelengths – insufficient to saturate the GPU using `GANG VECTOR` parallelism. Conversely, the horizontal dimension can

be tuned to ensure sufficient parallelism for `GANG VECTOR` parallelism, but results in non-coalesced memory accesses.

To evaluate alternatives, two strategies were tested in standalone versions of ecRad solvers. The first reorders the loops

so that the horizontal dimension is innermost, enabling efficient `GANG VECTOR` parallelism. The second strategy retains the

original layout and applies vector parallelism to the wavelength dimension and gang parallelism to the horizontal dimension.

Both strategies performed equally well in our benchmarks. Therefore, we selected the second approach, as it requires fewer

code modifications.

**Listing 3.** Example of an OpenACC block from the radiation component, with parallelization along `nproma`, `jc` loop, and `ng`, `jg` loop

```
1:      !$ACC PARALLEL DEFAULT(NONE) ASYNC(1) &
2:      !$ACC     NUM_GANGS(iendcol-istartcol+1) &
3:      !$ACC     NUM_WORKERS((config%n_g_sw-1)/32+1) VECTOR_LENGTH(32)
4:      !$ACC LOOP GANG PRIVATE(...)
5:      do jcol = istartcol,iendcol
6:          ...
7:          !$ACC LOOP SEQ
8:          do jlev = 1,nlev
```





```
 9:                    ...
10:               !$ACC LOOP WORKER VECTOR PRIVATE(...)
11:               do jg = 1,ng
12:                   od_cloud_new(jg) = od_scaling(jg,jlev) &
13:                      & * od_cloud(config%i_band_from_reordered_g_sw(jg),jlev,jcol)
14:                   ...
15:               end do
16:           end do
17:       end do
18:   !$ACC END PARALLEL
```

Note that the `jg` loop uses `WORKER VECTOR` parallelism instead of a standard `VECTOR` loop. This configuration has shown better performance with the OpenACC compiler used in our implementation.

### 3.7 Pollen

Pollen forecasts play a critical role in public health by enabling individuals with allergic conditions—such as hay fever and asthma—to anticipate high-exposure days, plan outdoor activities, and initiate preventive measures like early medication use. The modeling of pollen species in ICON is coupled with the aerosol module ART (Aerosol and Reactive Trace gases, (Rieger et al., 2015), (Schröter et al., 2018)) developed at the Karlsruhe Institute of Technology (KIT). In general, ART enables modeling a variety of trace gases and aerosols and their associated chemical processes. However, in the context of NWP production at MeteoSwiss, ART is used exclusively to simulate emissions, transport (see subsection 3.5), sedimentation, and washout processes of five pollen species during their blooming season, namely hazel, alder, birch, grasses, and ambrosia.

For production purposes, only the necessary ART subroutines required for modeling the above-mentioned pollen species are ported to GPUs. These porting efforts follow the same strategy as used for the ICON model (see subsection 3.2).

### 3.8 Data Assimilation

Data assimilation combines observational data with model output to generate the best possible estimate of the atmospheric state. This estimate, known as the "analysis", serves as the initial condition for subsequent numerical weather prediction (NWP) forecasts. Accurate and timely analyses are essential for high-quality short-range forecasts, particularly in data-rich regions like Europe.

Here we consider the so-called KENDA (Kilometer-scale Ensemble Data Assimilation) system (Schraff et al., 2016) used within ICON. It employs an ensemble Kalman filter, specifically the Local Ensemble Transform Kalman Filter (LETKF) by (Hunt et al., 2007), to assimilate a wide range of observational data. This includes conventional data from radiosondes, aircraft, wind radars and wind Light Detection and Ranging (LIDAR) instruments, and surface stations. Additionally, radar-derived surface precipitation rates are assimilated through a technique known as latent heat nudging.





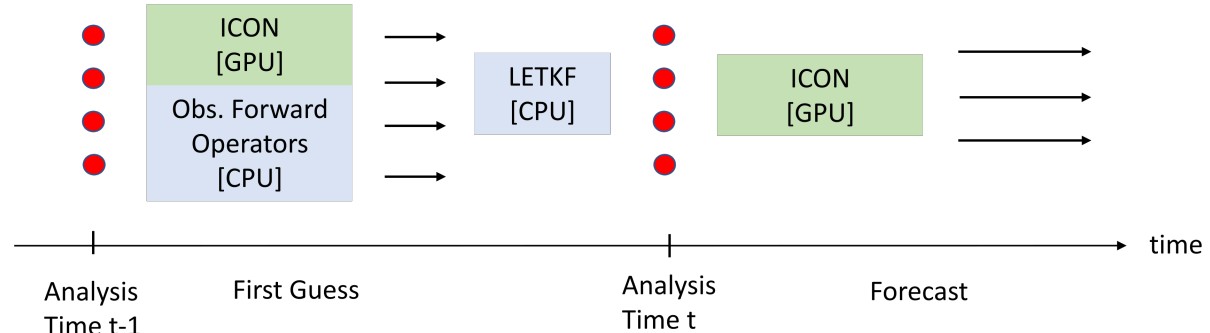

**Figure 3.** Diagram illustrating the GPU-CPU hybrid implementation of the ensemble KENDA assimilation cycle followed by a forecast.

The assimilation cycle consists of several stages, which can be seen in Figure 3. First, a one-hour ICON ensemble forecast
is run starting from the previous ensemble analysis time `t-1`, providing the so-called first guess. During this forecast, observation operators are applied to transform model variables (e.g., temperature, pressure, humidity) into the observational space and written to disk. The resulting innovations (observation-minus-forecast differences) are passed to the LETKF software to generate the new ensemble analysis at time `t`. This new analysis then provides the initial condition for the next forecast cycle. Observation operators pose unique challenges for GPU acceleration. In contrast to the structured grid-based computations
found in most of ICON, these operators work in the observational space, which is often sparse and irregular. This makes them poorly suited for execution on GPUs, which require high arithmetic intensity and regular memory access patterns for optimal performance. A detailed analysis of the computational patterns and data flow shows that for most operational configurations these operators are only called at very low frequency and only need a subset of the model state. Based on this, the following hybrid strategy is adopted: the first guess ICON run is executed on the GPU but the data assimilation observation operators are
kept on the CPU, and the required data are transferred from GPU to CPU during the run. The only exception is the so-called latent heat nudging of radar precipitation, which needs to be called at higher frequency and is therefore ported to GPU. The LETKF analysis step itself remains on the CPU.

### 3.9 Diagnostics and output

For operational NWP applications, a wide range of diagnostics must be computed to meet the needs of various clients and
users. These include, for example, wind gust statistics, lightning indices, and other derived meteorological products. Many of these diagnostics are evaluated at high temporal frequency, often every model time step, which makes GPU acceleration essential for performance. To minimize data transfer overhead, most high-frequency diagnostics have been ported to the GPU. This ensures that intermediate results remain on the GPU memory during the time-stepping loop, avoiding costly CPU-GPU transfers. At designated output intervals, the required diagnostic and model data are transferred from the GPU to the CPU.
Output-related operations such as file I/O are then performed on the CPU, which is more suitable for these latency-tolerant and system-dependent tasks. Pre-processing steps for output—such as interpolations to pressure levels, standard height levels, or



user-defined layers—are also ported to the GPU and executed before the data transfer. This further reduces CPU workload and ensures that only the final, ready-to-write fields are transferred from GPU memory.

## 4 Validation and acceptance

### 4.1 Probabilistic testing

To ensure the correctness of the GPU port of ICON, a probabilistic testing framework, probtest (Probtest, 2023), was developed. This framework is designed to validate scientific consistency between the GPU and CPU versions of the code while accounting for expected differences due to rounding errors and non-bit-reproducible floating-point behavior. Such differences commonly arise from hardware- and compiler-specific implementations of intrinsic mathematical functions or from numerical optimizations such as fused multiply-add (FMA) operations, which are applied aggressively on GPUs.

The key idea of probtest is to approximate the effect of rounding errors by constructing a CPU-based ensemble in which small perturbations are introduced into selected input fields—typically in the least significant digits. This generates a reference ensemble that captures the natural variability due to rounding effects in the CPU computation.

A short and computationally inexpensive configuration is used to run the perturbed ensemble and determine the expected spread of each variable. Based on this ensemble spread, variable-specific statistical tolerances are derived. The GPU simulation is then compared against this CPU-based ensemble, and the test passes if the GPU result falls within the ensemble spread for all diagnosed fields.

This probabilistic test is integrated into the automatic continuous integration (CI) pipeline, and is set up for the set of configurations which are supported on GPU using a reduced domain size. Every change to the model must pass this validation step to be accepted. In case of physical changes to the model that affect results beyond numerical noise, a new ensemble reference is generated to update the tolerance baseline.

### 4.2 Validation against observation

Due to the complexity of the model and to the fact that many conditional statements are data-driven, for example a cloud-no-cloud situation, it is not possible to have full code coverage with the above-mentioned automatic testing. In order to ensure the quality of the weather forecast, for every new version of a model that shall be used for production by a weather prediction center, an extensive validation, or so called verification, is required. To this end, the new version of the model is compared against observation for an extended period of time using different metrics such as Mean Error or Root Mean Square Error. Typically, the period of time can be a few weeks for the four seasons for past periods. Such verification has been carried out over multiple seasons for the MeteoSwiss configurations and is described in more details in Sect. 5.2. Such a verification should be repeated in case the model would be used for weather prediction using a different configuration.



## 4.3 Challenges of porting large community code

Ensuring the correctness of a GPU port for a large community is a critical step in integrating the port into the main code base. However, there is also a human aspect that should not be underestimated when integrating a major code extension that spans a broad range of components in a community-driven code like ICON.

Many ICON components, such as those described in Sect. 3.4 to 3.8, are developed and maintained by individual domain scientists (DSs), who are responsible for the scientific integrity and evolution of their respective modules. In contract, a research software engineer (RSE) ports a model to GPU by working across multiple code components without affecting the scientific value of the implemented equations. As a result, establishing trust between DSs and RSEs is essential, enabling DSs to understand the modifications required for the GPU port and maintain their ability to maintain and develop the ported code independently.

To facilitate this collaboration, we employ two strategies. First, ported code components are merged incrementally into the main code base. Second, DSs are trained in the basics of the GPU port, OpenACC, and tools for GPU verification, such as tolerance validation with probtest. In parallel, the RSEs gain familiarity with the scientific context of the code they work on, facilitating the interactions.

Incremental porting of the code ensures that the ported code remains up-to-date with the latest scientific advancements, simplifies testing and debugging, and allows DSs to continue working on their components immediately. Also, keeping the partially ported code in a working state enables early integration of GPU tests and probtest into the general ICON CI testing pipeline. Early GPU testing helps to avoid regression when a DS extends an already ported code component. To give a magnitude of the porting effort and the importance of an incremental approach, the source code is about 2 millions line of code, to which about 15 000 OpenACC statement have been added.

The GPU training provided for DSs and other RSEs working on ICON was well received by the ICON developer community, which comprises many scientists without a formal computer science background. The training was tailored specifically to ICON and the use of OpenACC within it. It was offered in live sessions and is also available for self-study, with many supporting documents, guidelines, and hands-on tutorials published and available to ICON developers.

While no formal evaluation was conducted, feedback from participants indicates that the training was perceived as helpful and played a positive role in promoting the acceptance of the GPU port and OpenACC-based development within the ICON community.

## 5 Operational NWP configuration at MeteoSwiss

MeteoSwiss currently runs two regional NWP ensemble systems, ICON-CH1-EPS and ICON-CH2-EPS, operationally. The ICON-CH1-EPS configuration has a horizontal grid spacing equivalent to 1 km and 80 vertical levels, with a time step of 10 s. It is run eight times per day, producing 33-hour forecasts for 11 members. The 03 UTC run is additionally extended to 45 hours to fully cover the next day. ICON-CH2-EPS uses a coarser grid spacing of approximately 2.1 km , with the same 80 levels and a 20 s time step. It runs four times per day providing 5-day forecasts for 21 members, offering a broader temporal range




while maintaining high spatial resolution for medium-range weather forecasting. In order to capture key weather phenomena for
Switzerland, the regional configurations are running on a simulation domain that includes the entire Alpine region as illustrated
in Fig. 4 such that ICON-CH1-EPS and ICON-CH2-EPS have, respectively, 1 147 980 and 283 876 horizontal grid points. The
initial conditions are provided by the KENDA-CH1 system, which combines a wide range of observations into the model
grid and physical equation. KENDA-CH1 has the same computational grid as ICON-CH1-EPS and is run for 41 members.
Lateral boundary conditions for these systems are supplied by ECMWF's IFS ENS system, which supplies high-quality global
atmospheric data.

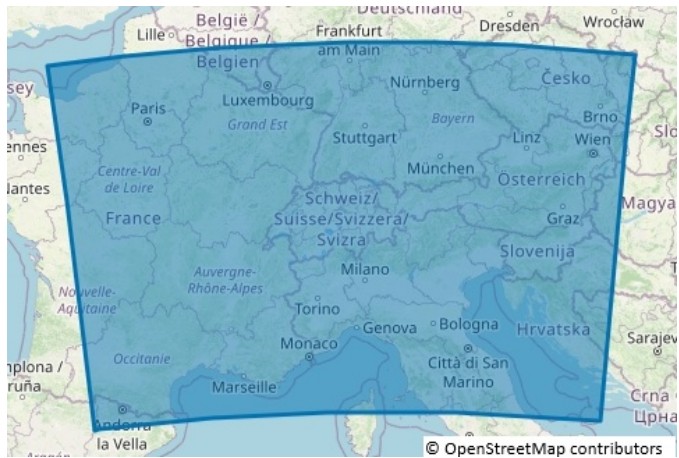

**Figure 4.** ICON-CH1-EPS and ICON-CH2-EPS domain covering the Alpine Region

## 5.1 Operational GPU System at CSCS

The MeteoSwiss computing infrastructure is part of the larger Alps Supercomputer at the Swiss National Supercomputing Cen-
tre CSCS (Alps, 2025) and is implemented as a so-called versatile cluster, vCluster (Martinasso et al., 2024), deployed across
multiple geographical sites for increased operational resilience. The production vCluster named Tasna is hosted in Lausanne
(western Switzerland), while the fail-over and R&D vCluster Balfrin is located in Lugano (southern Switzerland). vCluster
technology enables flexible configuration of the software environment and allocation of compute resources. Operational fore-
casting uses 42 GPU compute nodes, each equipped with one AMD 64-core EPYC CPU and four NVIDIA A100 96 GB GPUs.
These nodes are connected by a Slingshot-11 interconnect. The R&D size is of 42 compute nodes or larger depending on the
needs and in coordination with CSCS.

## 5.2 Verification of the MeteoSwiss configuration


After successfully ensuring the consistency of the GPU port against the CPU execution with automated testing (see Sect. 4),
an extended verification of the ICON model was conducted. Before introducing it into operations, the forecast quality of
the new system was thoroughly assessed and compared to the previous operational system that was based on the COSMO





model (Steppeler et al., 2003; Baldauf et al., 2011). To this end, MeteoSwiss ran ICON-CH1-EPS and ICON-CH2-EPS control
forecasts on a regular schedule (00 and 12 UTC runs) starting in summer 2021, transitioning to full ensemble runs in summer
2023. For the first period the model was run on CPU, and from November 2022 on it was run on GPU.

Figure 5 presents an extended verification of ICON-CH1-EPS against the then-operational COSMO-1E system. Since sum-
mer 2023, when ICON ensemble data became available, the median of ICON-CH1-EPS consistently outperforms COSMO-1E
in terms of the equitable threat score (ETS) for 12-hourly precipitation exceeding 0.1 mm, indicating greater skill in capturing
the occurrence of precipitation. Moreover, the mean absolute error (MAE) for 2-m temperature and 10-m wind speed is slightly
lower since summer 2023, suggesting an improved overall forecast accuracy for these surface parameters. The only exception
is total cloud cover, where the ETS for the 2.5 octa threshold shows slightly lower performance for ICON-CH1-EPS in recent
seasons.

Given the slightly improved performance of ICON-CH1-EPS (and ICON-CH2-EPS; not shown) in many of the important
parameters (see some of them in Fig. 5), the quality criteria for operational introduction were met and ICON replaced COSMO
as the operational NWP system at MeteoSwiss on 28 May 2024. Nevertheless, several known model deficiencies remain,
including a warm bias in Alpine valleys, an overestimation of convective precipitation maxima, and an excessive ensemble
spread in precipitation forecasts. These deficiencies are the focus of ongoing development with the goal to further enhance the
quality of the weather forecasts.

## 6 Optimization and performance results

### 6.1 Benchmark configuration

The benchmarking experiments presented in this section are based on the operational ICON-CH1-EPS configuration (see
Sect. 5), using a single ensemble member for a one-hour forecast. This choice is representative of a forecast simulation, par-
ticularly considering that diagnostic and output components are typically called at hourly intervals, while most other model
components are invoked at sub-hourly frequencies. The exact configuration namelist can be found in the open source ICON
release (https://www.icon-model.org/, release icon-2024.10) under the test name `mch_icon-ch1`. Each benchmark measure-
ment was repeated ten times, and the mean elapsed time and its standard deviation were recorded. The ICON version used for
the benchmarking is based on the release-2024.10, on top of which some of the optimization discussed below have been added.
Most of the optimizations are part of the official public release-2025.4 of ICON (Icon-model, 2025).

All experiments are run on the Balfrin hybrid system (see the specification in Sect. 5.1). Unless stated otherwise, benchmarks
are executed on two GPU nodes, e.g. using a total of 8 NVIDIA A100 96 GB GPUs. The optimization parameter controlling
the block size for the horizontal grid `nblocks_c` is set to 1, and the radiation block parameter `nproma_sub` is set to 6054.
The code was compiled using the NVIDIA HPC SDK version 23.3 with CUDA version 11.8.0.





**Figure 5.** Extended verification of COSMO-1E and ICON-CH1-EPS forecasts for the lead-time range of 13–24 h, evaluated against observations from 159 surface stations across Switzerland. Shown are key meteorological verification scores: (a) ETS for 12-hourly precipitation > 0.1 mm, (b) ETS for total cloud cover > 2.5 octa, (c) MAE of 2-m temperature, and (d) MAE of 10-m wind speed. Each panel presents individual seasonal score values (points) and a moving yearly average (lines), computed from the current and the three preceding seasons, for COSMO-1E (blue) and ICON-CH1-EPS (orange). Scores are based on the ensemble median, except for ICON-CH1-EPS prior to JJA 2023, where ensemble data were not yet available. The model was run on GPU from November 2022 on, the results in the previous periods have been obtained on CPU.





## 6.2 GPU optimization

After porting and validating the ICON code on GPUs, a range of optimizations have been introduced to improve performance. These optimizations are described below, and the resulting performance of the combined optimization is shown in Table 1.

### 6.2.1 Baseline

The GPU timings are compared to a CPU reference. Since some of the optimizations also improve the CPU runtime, we use the best-performing optimized version as the CPU reference, including mixed precision and radiation in single precision. The CPU reference is compiled with the following optimization flags `-O -Mrecursive -Mallocatable=03 -Mbackslash` and is run using 8 AMD 7713 64 cores EPYC Milan CPUs, using MPI parallelization over all cores. The parameters for the horizontal blocking are set to $nproma = 8$ and $nproma\_sub = 8$ which are optimal values on CPU for this configuration. The total time for this 1 h-benchmark is 536.5 s. Note that due to an issue with the nvhpc compiler available on the system, the hybrid OpenMP - MPI parallelization did not work on the CPU for this configuration and is not reported in the table 1. The base GPU code is compiled with `-O -Mrecursive -Mallocatable=03 -Mbackslash -acc=verystrict -Minfo=accel,inline -gpu=cc80`. Comparing the CPU reference on 8 CPU sockets with the GPU code running on 8 A100 GPUs, we observe a total runtime of 125.6 s and a speed-up factor of about 4.3. Note that we favor this socket-to-socket comparison as opposed to a node-to-node comparison since the GPU nodes have more GPUs than CPUs which would be too favorable for the GPUs. Since the weather model ICON is mostly memory bandwidth limited (Adamidis et al., 2025), it is instructive to compare the speedup observed with the hardware specification. The NVIDIA A100 96 GB and the AMD 7713 EPYC CPU have a maximum theoretical bandwidth of, respectively, 1560 GB/s and 204.8 GB/s, which gives a ratio of 7.8. This is consistent with the observed speedup of a factor of 4.3, which suggests that the initial port is performing reasonably well and that no significant computations have been left on the CPU. But it also indicates that there is potential room for improvement.

### 6.2.2 Compiler optimizations level and flags, comp-opt

First, various optimization flags at the compiler level have been investigated and compared. Aggressive optimization flags such as `-O3` or `-fasthmath` have not been considered for accuracy and validation reasons. Using `-O2` and `-Mstack_arrays` noticeably reduces the total runtime by about 6.8% to 117.0 s Most of the gain comes from the flag `-Mstack_arrays` which places all temporary Fortran automatic arrays on the stack. Although this change only affects memory allocation on the CPU, it also impacts GPU runtime when using OpenACC, because these automatic arrays are still allocated, even if never used on the CPU. The effect is, in fact, larger for the GPU run, because of the very large `nproma` used (see Sec. 3.2), which leads to large temporary arrays.





### 6.2.3 Asynchronous execution between CPU and GPU

By default, OpenACC synchronizes CPU and GPU execution, such that the CPU needs to wait for the GPU kernels to complete
before being able to proceed. This can be adapted by using the OpenACC `ASYNC(INT)` constructs in parallel regions, allowing the CPU to continue execution after launching the kernel. Asynchronous computation needs a careful analysis of the code to ensure that no data computed on the GPU are used at a later stage on the CPU, while computation is still ongoing on the GPU. In such a case, the construct `ACC WAIT(INT_LIST)` is used to wait for completion of the GPU execution before proceeding. The OpenACC `ASYNC` construct further allows the specification of the queue – corresponding to the cuda-stream on NVIDIA
devices – where the asynchronous execution is done. In most of the ICON code, the asynchronous queue is explicitly set to 1, except for specific parts where multiple code paths are executed in parallel queues. This is, for example, the case for the different tiles of the soil as described in Sect. 6.2.5. With asynchronous execution, the CPU can proceed and launch multiple kernels, significantly reducing or even eliminating kernel launch overhead. As shown in table 1 this brings down the total runtime to $115.3\,s$, i.e., about 1.5% additional performance improvement.

### 6.2.4 Inlining

Inlining is a general optimization technique that replaces the call to a function by the actual code of the function. This can reduce function call overhead and allows for additional compile-time optimizations. It is particularly beneficial for functions which are called from the innermost-loop. In Fortran there is no language construct, as in C for example, to control inlining. However, most compiler vendors provide solutions to inline functions. With the nvhpc (NVIDIA) compiler there is a pre-
compilation step added where the user gives a list of functions that should be inlined, such that these functions are extracted as code into an inline library. This inline library is then used for the compilation of the full code. Inlining is not always beneficial and significantly increases the compilation time. For this reason, inlining is restricted to functions resulting in a significant performance improvement. In ICON, these are primarily in the physics scheme. The total runtime reduces to $113.2\,s$, which corresponds to a relative improvement of 1.8%.

### 6.2.5 CUDA Graphs

CUDA Graphs are an NVIDIA-specific GPU optimization feature that allow GPU workloads to be expressed as a directed acyclic graph (DAG) of operations, rather than launching individual kernels sequentially. This enables the GPU runtime to schedule and execute kernels with significantly reduced launch overhead and memory allocation costs. In ICON, CUDA Graphs are employed within selected physical parameterizations, most notably in the turbulence and soil components of the NWP
configuration. In addition to the turbulence transfer and the soil model, the CUDA graph is used in association with multiple asynchronous queues to run all the independent soil type tiles concurrently in different GPU queues.

At runtime, all GPU kernels, memory allocations, and de-allocations with their respective dependencies between the OpenACC extension APIs `accx_begin_capture_async` and `accx_end_capture_async` are recorded as a graph. After





that, the recorded graph can be executed on the GPU via the `accx_graph_launch` API multiple times, which is called a graph replay.

Although the required code changes are minimal, their use introduces certain limitations, which may require further adaptations to the code. The most critical aspect is that, during graph replay, no CPU work should be carried out; only prerecorded GPU kernels should be launched. In addition, all OpenACC statements within the capture region must be asynchronous. Moreover, kernel parameters are captured by value during the recording and cannot be changed later in the execution. This means that all array shapes and pointers cannot change from one graph replay to another. In ICON some fields have two time levels, so that the pointers associated with such fields are different for odd and even time steps. Therefore, for the part of the code that uses fields with two time levels, two graphs need to be recorded, one for odd and one for even time steps.

The application of CUDA Graphs to the soil and turbulence routines yields significant local speedups ($3\times$ and $3.5\times$, respectively). While these routines constitute a relatively small portion of the total runtime, the aggregate benefit of CUDA Graphs translates to a 5% overall runtime reduction, bringing the benchmark execution time down to 107.4 s.

### 6.2.6 opt-rank-distribution

When running the model on multi-GPU nodes, there is one MPI task associated to each GPU. In addition, some of the remaining CPU cores on the node are used for I/O related tasks, namely asynchronous I/O MPI tasks and a pre-fetch MPI task. Pure CPU MPI tasks should be evenly distributed over all nodes. Since I/O related tasks are at the end of the MPI communicator, they can be distinguished from compute tasks in a straightforward way by using the slurm `SBATCH –cyclic` command. However, this has the drawback that close MPI ranks are on different nodes, which could mean close domains are on different nodes, leading to more inter-node communication. An optimization of the rank distribution (opt-rank-distribution) can be achieved using `SBATCH –distribution=plane=4`. With this change, the total time improved by 1% to 106.3 s.

### 6.2.7 Compile-time nproma

The parameter `nproma` is usually set during runtime via a namelist switch. The `nproma` is chosen for the best performance dependent on the computing architecture. In many architectures, this is fixed to accommodate for the cache size or vector length. For GPU, the `nproma` is chosen to be as large as possible and dependent on the number of grid points. Setting the value at compile time provides a performance increase. Although this optimization is available, it is currently not used for the operation at MeteoSwiss since the two configurations ICON-CH1-EPS and ICON-CH2-EPS have different numbers of points and therefore would require two different executables with different `nproma`. With this optimization, a gain of 0.7% is obtained corresponding to a total runtime of 106.3 s.

### 6.2.8 Dycore mixed-precision

Using single precision can give multiple advantages over double precision. On the one hand, more floating point operations can be done in the same time as double precision operations. On the other hand, it decreases memory pressure, which is useful,





especially considering the limited amount of memory available on the GPU. The drawback of using single precision is that the errors of using floating point numbers increase, and the effect on the numerical integration need to be carefully analyzed. For this reason a mixed-precision approach has been implemented in ICON's dycore. Scientific expertise determines which fields and operations can be put in single precision. The mixed-precision model was validated against a double-precision version by comparing to observations over a long period of time, which showed no impact on the meteorological scores. The performance
is improved by 8% to 96.65 s.

### 6.2.9   ecRad single-precision

Similar considerations as for the dycore have been made for the radiation parameterization ecRad regarding the floating-point precision. We also note that ecRad is already used in single-precision for operations at the ECMWF in combination with the weather model IFS which provides additional validation to this approach. With this change, the total runtime is improved
by 0.9% to 95.80 s. The ecRad parameterization is a memory-intensive component such that using single-precision has a significant impact on the total memory consumption. This could be beneficial for the overall performance since it allows to increase `nproma_sub` to use more parallelism and reduce launching overhead.

    Looking at the overall improvement, the benchmark runtime was optimized by 23% from 125.6 s to 95.80 s. Comparing to the CPU reference at 536.5 s this gives a final socket-to-socket speedup of 5.6x on the GPU. This speedup factor is consistent
with previous results for climate configuration setup (Giorgetta et al., 2022). The speed up factor also means that compared to a CPU only a GPU system for operation is more compact, 5.6x time more CPUs would be needed for the required time to solution. Although we did not perform energy measurement in this work, a previous study (Cumming et al., 2014) considering a similar model, shows that the better performance also translates in a more energy efficient system.

### 6.3   Strong scaling

To assess strong scaling behavior, the ICON-CH1 benchmark is run using an increasing number of GPUs. The reference time is given for 4 GPUs, which is the smallest number on which the problem fits in terms of memory. The most optimized version from the previous Sect. 6.2, which is the ecRad-single-precision version, is used for this test. In Figure 6, it can be seen that the problem scales well up to 12 GPUs.

    Beyond 12 GPUs, there is a noticeable degradation in performance in the physics component, while the dynamical core (dycore)
continues to scale well. Considering that the ICON-CH1-EPS grid has 1147980 grid points, with 12 GPUs the number of grid points per GPU is below 100000. At this point, the per-GPU workload becomes too small to fully utilize the compute and memory throughput capabilities of the NVIDIA A100 GPUs. In particular, the ability to overlap memory access with computation is reduced, leading to under-utilization. In particular, we note that there is little communication in the physics as compared to the dycore, which further supports that the non-optimal scaling results from non-optimal use of the GPUs rather
than communication overhead.




| Name | Dycore | | | Physics | | | Total | | |
|---|---|---|---|---|---|---|---|---|---|
| | Mean [s] | Std [s] | Speedup | Mean [s] | Std [s] | Speedup | Mean [s] | Std [s] | Speedup |
| CPU ref | 421.8 | 5.168 | 1. | 135.7 | 1.754 | 1. | 536.5 | 1.535 | 1. |
| base GPU | 64.53 | 0.2718 | 6.537 | 40.41 | 0.2821 | 3.359 | 125.6 | 0.5964 | 4.271 |
| comp-opt | 63.19 | 0.09597 | 6.675 | 36.05 | 0.1576 | 3.765 | 117.0 | 0.1766 | 4.584 |
| async | 62.01 | 0.09919 | 6.802 | 35.67 | 0.07311 | 3.805 | 115.3 | 0.1222 | 4.652 |
| inlining | 62.03 | 0.01604 | 6.800 | 33.28 | 0.02778 | 4.078 | 113.2 | 0.03935 | 4.740 |
| cuda-graphs | 62.06 | 0.09837 | 6.796 | 27.68 | 0.1052 | 4.904 | 107.4 | 0.06748 | 4.995 |
| opt-rank-node | 61.29 | 0.05912 | 6.883 | 27.70 | 0.2921 | 4.899 | 106.3 | 0.09753 | 5.048 |
| compile-time nproma | 61.00 | 0.08110 | 6.915 | 27.54 | 0.07775 | 4.928 | 105.6 | 0.07021 | 5.079 |
| mixed-precision | 52.68 | 0.1008 | 8.007 | 27.55 | 0.2176 | 4.927 | 96.65 | 0.2031 | 5.551 |
| ecRad-single-precision | 52.76 | 0.1240 | 7.996 | 26.73 | 0.5259 | 5.078 | 95.80 | 0.2762 | 5.600 |

**Table 1.** Performance metrics for different parts of the model with various optimizations. For the speedup reference the CPU version is used, which includes all the optimizations that also affects the CPU performance like CPU optimization flags, in-lining, opt-rank-node, mixed-precision and ecRad in single precision.

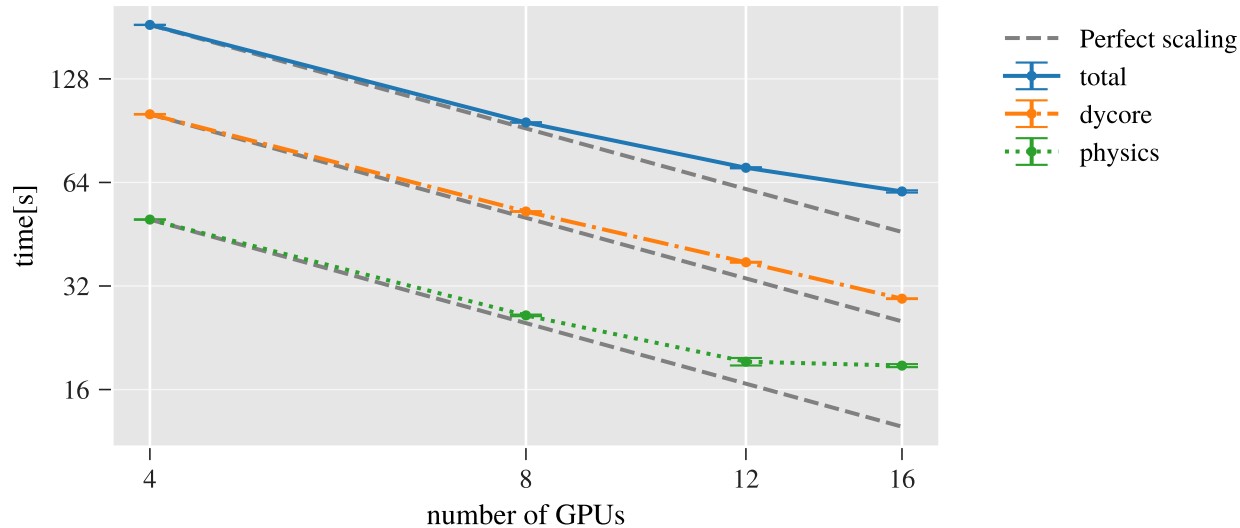

**Figure 6.** Strong scaling of the regional ICON-CH1-EPS 1h benchmark on A100 NVIDIA GPUs. For this configuration the model stops scaling at about 12 GPUs which corresponds to about 100 000 Grid points per GPUs and is not enough work to optimally use the hardware.





## 6.4 Timing for the operational configuration

For the operational ICON-CH1-EPS configuration at MeteoSwiss, each 33-hour ensemble forecast must be completed within 50 minutes (3,000 seconds) to meet the stringent timing requirements for delivering downstream critical products to clients. When ICON was first deployed operationally in May 2024, not all GPU optimizations were yet in place. At that time, the runtime on two nodes (8 NVIDIA A100 GPUs) was approximately 3,200 seconds, exceeding the operational time limit. As a result, each ensemble member had to be executed on three nodes (12 GPUs) to stay within the required timeframe. After applying the full set of optimizations, the runtime for a 33-hour forecast on two nodes was reduced to 2,642 seconds, well within the operational constraint, enabling more efficient and cost-effective use of computational resources.

## 7 Conclusions

We have successfully ported all essential components of the ICON model required for operational numerical weather prediction (NWP) to GPUs using OpenACC compiler directives. This includes the dynamical core, all physical parameterizations, and the data assimilation system (KENDA). The porting and optimization strategy has enabled a single ensemble member of the operational ICON-CH1-EPS configuration to run within the required time-to-solution on 8 NVIDIA A100 GPUs.

When comparing the same number of CPU and GPU sockets, the speedup on the GPU hardware is 5.6x. This potentially allows for a much more compact and energy-efficient system than a CPU-only system. The OpenACC approach enabled the implementation of changes in a step-by-step manner in the code while maintaining performance on other architectures. Besides the MeteoSwiss configuration, several other configurations are also supported and tested including one configuration from Germany's National Meteorological Service, the Deutscher Wetterdienst (DWD).

All changes have been merged into the main ICON code and OpenACC training in ICON was provided to the community to promote a seamless adoption of the port and would allow other NWP centers or users to run ICON on GPU. The code changes are included in the ICON open-source distribution, with the exception of the data assimilation part, making it available for many use cases to the broader NWP community. The ICON model was thoroughly verified over multiple seasons against observations and the previous operational model COSMO achieving the required quality level for the MeteoSwiss configurations. The port allowed MeteoSwiss to be the first weather service to use the ICON model on the GPU in production for operational NWP prediction. Thanks to the computing efficiency of GPUs, MeteoSwiss can run, in particular, the ICON-CH1-EPS configuration at up to 1 km resolution, which remains one of the highest-resolution ensemble systems to date.

Beyond immediate operational gains, the GPU-capable ICON model is well positioned to benefit from the growing convergence of physics-based and machine learning (ML) approaches in weather and climate modeling. Shared GPU infrastructure can now support both traditional simulations and ML-based training and inference workflows, enabling hybrid strategies central to next-generation forecasting systems.

Nonetheless, the OpenACC approach comes with limitations in terms of maintainability, optimization and portability. Currently, the OpenACC standard is only fully supported by the nvhpc compiler on NVIDIA hardware. Running ICON with AMD GPUs is, for example, currently only possible on a Hewlett Packard Enterprise (HPE) system using the Cray compiler for



climate configuration, while there are some unresolved issues with some NWP components on such system. Finally, there are
no solutions for running OpenACC code on Intel GPUs. To improve portability and performance on GPU hardware, the ICON
community is investigating several other approaches, including a rewrite using the Python domain-specific language GT4Py
(Paredes et al., 2023).

*Code and data availability.*  The full ICON version needed for all production configurations at MeteoSwiss, including the closed source Data
Assimilation components for the KENDA cycle, used in the paper is archived on the Zenodo server (https://zenodo.org/records/15674269)
under DOI (ICON, 2024.10_withDA). This code version is available under restricted access for review and for research purpose. For the icon-
ch1-eps benchmark used for the performance results reported in the paper, the data assimilation component is not needed, the benchmark
results can be reproduced using the official open-source release source code, configurations and scripts available under BSD-3 License
archived on the World Data Center for Climate server from DKRZ (https://www.wdc-climate.de/ui/entry?acronym=IconRelease2024.10)
under DOI (ICON, 2024.10). A copy of the input data required to run the ICON-CH1-EPS benchmark as well as the raw data used for
the verification for the years 2021-2024 is available on the Zenodo server (https://zenodo.org/records/16760638) under DOI (ICON-CH1-
EPS_data).

*Author contributions.*  XL supervised and coordinated the GPU porting effort and ported part of the code to OpenACC. DH ported several
components to OpenACC in particular the radiation scheme and optimized the code. FG ported components of the model in particular the
data assimilation to the GPU and worked on optimizations. AL worked on the testing infrastructure and ported components to OpenACC. AW
was the project manager for the operationalization of ICON at MeteoSwiss and coordinated the effort between the GPU port and operational
requirements. CM worked on the optimization of the dycore. DA worked on many optimizations in particular in the dycore and implemented
the CUDA Graph solution. JJ ported part of the code, in particular some diagnostics to the GPU. MaJ ported several components in particular
those required for the DWD operational NWP setup. MiJ ported components required for regional climate modeling setups. MS ported the
pollen module and worked on the optimizations. RD ported components, in particular the microphysics and turbulence parameterizations
and developed the probtest infrastructure. US optimized the convection scheme. VS ported the subgrid scale orography parameterization.
WS ported many components, in particular the dycore and infrastructure such as the communication. CM and DL contributed to the Data
Assimilation system. PB prepared the system for operation. CO supervised part of the work. MA supervised the improvement of the model
configuration at MeteoSwiss. LJ and DL optimized and improved the model configuration. PK and AP worked on the verification of the
model. OF and PS supervised part of the work and operationalization of the system. MK and MI worked on the computing system to
make it ready for operation. RA and TS supervised the preparation and project of the operational computing infrastructure. BC prepared
the operational software stack. SK supported the implementation of the build system. RM supported the implementation of the CI testing
infrastructure for the operational system.

*Competing interests.*  Author Dmitry Alexeev is employed by NVIDIA. The other authors have no competing interest



*Acknowledgements.* The authors would like to thank the ICON partners MPI-M, DWD, DKRZ, and C2SM, as well as the ICON developers,

for their support and, in particular, for their help with reintegration into the main code, resolving merge conflicts, and addressing various issues. Special thanks go to the German computing center, DKRZ, for providing the Buildbot CI testing infrastructure, which was key to the success of this project. The authors would also like to thank the CSCS engineers for their excellent support and for ensuring the system's success and stability. The GPU port of ICON for NWP application was coordinated as part of the priority project IMPACT of the COSMO consortium. The work was partially funded by the C2SM,ETHZ-MeteoSwiss project Weiterentwicklungen-ICON (WEW-ICON).

The authors acknowledge the use of the AI tool DeepL Write for improving the style of some sentences in the manuscript. This paper is dedicated to our beloved colleague, André Walser, who passed away in February 2025. He played a leading role in bringing the ICON model into production at MeteoSwiss.



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
