# Peer review of "Operational numerical weather prediction with ICON on GPUs"

_EGUsphere, 2025_

## Author Response (AR1)

**Reply to Anonymous Referee #1**

The authors would like to thanks the referee for the careful analysis and comments of the manuscript. A detailed answer can be found below.

> The introduction lists several other efforts to adapt existing weather and climate models to GPUs but leaves out DSL-based efforts that aim at providing efficient CPU and GPU support through several code generation backends. This includes, e.g., PSyclone/LFric (UKMO/STFC) or GT4py-based models, such as ICON-EXCLAIM (MeteoSwiss/ETHZ), PMAP (ECMWF/ETHZ), PACE (AI2). Although this serves more purposes than GPU porting, the performance portability to heterogeneous hardware is a guiding principle for such approaches.

Citations for the various efforts using DSL approach is added in the introduction : "Other models have or are beeing re-written using a Domain Specific language (DSL) approach, allowing to potentially reach a better performance portability as well as a separation of concerns between the user code and hardware optimizations.

The COSMO (Consortium for Small-scale Modeling) model was ported using a combination of DSL rewrite and OpenACC directives, achieving substantial speed-ups on GPU systems with respect to the CPU baseline~\citep{lapillonne2014, fuhrer2014}. It was also the

first model used by a national weather service for operational numerical weather prediction on GPUs. The LFric model developed by the UK MetOffice is using the Psychlone DSL~citep{adams2019lfric}. Several ongoing effort are considering the GT4py (Grid Tools for python) DSL this includes the Pace model \citep{Dahm2023} from Allen Institute for Artificial Intelligence (AI2), the Portable Model for Multi-Scale Atmospheric Predictions (PMAP) model~\citep{Ubbiali2025} at ECMWF, or the ICON (Icosahedral Nonhydrostatic) model~\citep{Dipankar2025}. Recently the Energy Exascale Earth

System (E3SM) was entirely rewritten using C++ and the Kokkos library~\citep{Donahue2024}.

."

>In terms of effort, the fact that weather and climate models often comprise millions LOC has been mentioned prominently but no indication is given into what order of magnitude ICON falls or what share of ICON had to be adapted additionally for the use at MeteoSwiss.

The following text is added in section 3:

"A simple code analysis indicates that the ICON model consists of approximately 1.2 million lines of Fortran code, excluding comments and non-executable directives, with an additional

21,000 OpenACC pragmas introduced for GPU acceleration. Restricting the analysis to the subfolders used in operational numerical weather prediction (NWP), it reduces these figures to around 660,000 and 13,000 lines, respectively."

is properly placed.

>The description of ICON's parallelization is very handwavy and statements such as "advanced data structures and efficient communication protocols ensure scalability" would be more believable with some accompanying words what makes these advanced and efficient or references for this.

The sentence is reformulated to be more precise and a citation added "The communication accross nodes is implemented using the Message Passing Interface (MPI) library and the model is scaling well up to thousand nodes (Gioregetta, 2022).

The vertical discretization in ICON is based on a terrain-following hybrid

coordinate system which combines the advantages of pressure-based and

height-based coordinates."

> Listing 1 is presumably intended as a first glimpse at how the OpenACC code looks like, but it references some concepts that are (partially) explained only much later, such as nproma, async, or the OpenACC parallelization loop annotations in general. Since the text explains other OpenACC concepts later, it would be useful to readers without knowledge of the directives to briefly describe what these comprise of, how they are mapped to the GPU hardware in terms of parallelization/execution model, and data movement/availability.

This listing is serving as a first example of OpenACC, Listing 1 is replaced by a simple addition operation which do not require introducing ICON terms.

> In ll. 85-86, the authors claim that low arithmetic intensity is responsible for the fact that porting isolated kernels to GPU would offer little benefit. I don't agree with this causal link for two reasons: (1) I consider arithmetic intensity a useful metric that allows to estimate the potential for performance benefits from GPU execution of a code, but because GPUs do not only excel in executing large amounts of floating point instructions but also provide a much higher memory bandwidth than most CPU architectures, also low AI codes can benefit from GPU execution. (2) In my opinion, the question whether porting of individual kernels can be sufficient for performance improvements is mostly related to the performance profile of the full application. A flat performance profile with multiple kernels that contribute similar runtime shares makes it difficult to create measurable performance gains, while an application with a

dominant kernel may benefit from just porting that one kernel if it has the potential to gain from GPU execution.

The initial formulation was not clear. We did not meant that the low intensity is an issue for the performance of the individual kernels,  but rather that if not all computations using 3d fields are ported to GPU, the time will be dominated or at least very impacted by data transfer such that there will be little speedup overall. We propose the alternative formulation:

"In atmospheric models, the arithmetic intensity, i.e., the ratio of computation (floating-point operations) to memory accesses, is generally low (Adamidis et al., 2025). As a result, only porting isolated kernels to the GPU would incur a significant penalty of data transfer between the CPU and the GPU that cannot be amortized by the ported kernel execution time"

> In the explanation of the loop collapse (ll. 116ff.), I would think it would be useful to relate the impact of the loop collapse to the SIMT execution concept of GPUs. In combination with the fast switching of lightweight threads, this makes it intuitively clear why more parallelism is so essential on GPUs to keep execution units occupied - a fact that is often overlooked by developers that are more familiar with SIMD execution models.

The text is reformulated as follow to make it clearer:  "The inner nproma-loop can be collapsed

with the outer nlev loop when appropriate, singnificantly increasing the amount of available parallelizm. This allows to use larger kernel grids that, typically, improves average occupancy and thus helps better hide memory latency. "

> In l. 133: "generally consistent across most NVIDIA architectures": I think "across most NVIDIA architecture generations" is what is meant here?

This is corrected accordingly

> In the discussion of the radiation scheme, it is not clear to me what is meant by the "reduced grid": Does ICON perform radiation computations on a lower resolution grid, or is this a reference to the reduced number of gridpoints within an nproma block? The description of the nproma_sub blocks is also somewhat repeated between the two paragraphs in ll. 193 to 206. I would advise some editorial changes to streamline the presentation in this section.

A definition is added: "radiation

calculations are performed on a coarser grid, so called reduced grid, "

> ll. 231-232 claims that WORKER VECTOR performed better than standard VECTOR loops but gives no evidence why that might be the case. Does this result in a difference in the launch configuration/block size that the NVIDIA runtime chooses for this particular loop? Or is there a different explanation?

After further investigation this is in fact related to a limitation of the compiler version used for this work and is not a general statement. The text is adatped as follow:

"This configuration addresses compiler limitations and ensures proper utilization of the desired number of CUDA threads."

> The porting strategy in Section 3 focusses exclusively on computational aspects but excludes the management of managing data buffers on CPU and GPU and the implementation of any data transfers between these. Given that A100 GPUs are the target platform, it is unlikely that a unified memory model is used, but is there use of managed memory or are all GPU allocations and transfers explicit via OpenACC data directives? Is there any use of automatic allocations in acc routines or are any other means in place to improve memory handling, e.g., via the use of pool allocators?

The description of the memorAll data management and allocation is explicit using openacc statement. There are is no pool allocator in the code, but the compiler may use such implementaion in the background in particular for fortran automatic arrays in subroutine. The Nvidia compiler is doing such optimizaiton which is in fact key for the performance since it avoids recurent allocations inside those subroutines.

> The probabilistic testing method described in Sec. 4.1 seems similar to the Ensemble Consistency Testing methodology used by CESM and MPAS. How does the probtest method used here differ from this approach?

The probtest method shared some similarities with Consisteny Testing methodology, but is somewhat more tailored towards validation of GPU port or hardware opitmizations, in particular with the probtest approach we do not exclude any variables, but rather compute a tolerance for each variable. The probtest method is based on the testing infrastucture that was developed for the port of the COSMO model to GPUs as described in "Using Compiler Directives to Port Large Scientific Applications to GPUs: An Example from Atmospheric Science", PPL Lapillonne, 2014. The authors believe that a details description of the possible testing method might be beyond the scope of this paper.

> I like Section 4.3 that discusses some of the challenges related to porting a community code and the role of domain scientists. It does not become clear, however, how the feedback and adoption was from this less technical group of developers: Even with additional training, were subsequent code contributions into already ported components always suitable for GPU execution? Or do they sometimes/often require further adaptation? Moreover, with OpenMP and OpenACC directives in the same code base, readability of the code often suffers. Is this an issue, particularly with domain scientists?

Yes for simple development, most developers are able to extend already ported code. The following sentence is added:

"Finally thanks to the training most ICON developers are able to contribute further changes and additions to ported code. For more complex implementation support from expert GPU developers is provided."

>The description of the GPU system in Sec. 5.1 was not sufficiently clear to me, in particular what the 42 GPU nodes resemble: Is this the number of nodes required for one ensemble member (I don't think so), or the number of nodes required to run all ensemble members in parallel, or simply the size of the production cluster? If the latter, how many nodes are in use in total when producing an ensemble forecast?

The following sentence "Each member of ICON-CH1-EPS is running on 3 nodes ..." is added.

> Is my understanding correct that the benchmark configuration in Section 6.1 does not include any I/O (nor Data Assimilation)? Since these parts are CPU resident (per Fig. 1/Fig. 3), it would be interesting to also have an assessment of the additional cost incurred by the necessary device-to-host data transfers, and whether any work has been done to optimise these - e.g., via the use of pinned memory buffers.

The benchmark does includes I/O and the part of data assimilation that is used in ICON-CH1, e.g. latent heat nudging. All corresponding data transfer are included in the Total time. The following sentence was added "The reported timings includes all data transfers..."

> Section 6.1 does not explicitly state the compiler choice (and which version) - for OpenACC execution the obvious choice is NVHPC but it becomes only implicitly clear that this is likely also the compiler used for the baseline results on CPU.

The sentence was clarifed adding the word compiler : "The code was compiled, both for CPU and GPU ..."

> Also, note that optimization flag "-O" (in contrast to "-O2") disables SIMD vectorization, which tends to improve safety towards floating point exceptions but may significantly limit performance on modern CPUs and thus potentially result in an artificially reduced baseline performance, particularly since GPUs will still use FMA instructions etc:

From the NVHPC man page:

    -O      Sets the optimization level to 2, with no SIMD vectorization enabled.  All level 1 optimizations are performed. In addition, traditional scalar optimizations such as induction

            recognition and loop invariant motion are performed by the global optimizer.

    -O2     All -O optimizations are performed. In addition, more advanced optimizations such as SIMD code generation, cache alignment and partial redundancy elimination are enabled.

The reference was re-computed with -02 which is slighly faster. All the speed up numbers have been adapted accordingly. It changes the overall speed up from 5.6x to 5.5x

> The vague mentioning of "an issue" with OpenMP-MPI is also somewhat unsatisfactory, could this be further elaborated and whether this is something that has been brought up with the vendor or was resolved in a later compiler version? Lastly, the given bandwidth in l. 406 is presumably the peak bandwidth of the main memory?

A bug report was submitted to Nvidia. The issue should be fixed in the latest version of the compiler but requires some adaptation on the ICON code. The latest version of the nvidia compiler is not available on the MeteoSwiss system. We note however that our speed up value are in agreement with results from Giorgetta 2022 where some of the ICON CPU reference results were obtained using OpenMP on a different system. The following is added :

"The problem was reported to Nvidia, and shall be adressed in later release of the compiler.

> Lastly, the given bandwidth in l. 406 is presumably the peak bandwidth of the main memory?

Yes, the following is added " a maximum theoretical

bandwidth to the main memory of"

> Out of curiosity: Do "-O3" or "-fastmath" change the accuracy of the results in a way that is picked up by probtest? Or does it change them to unphysical values?

-O3 and -fastmath did not validate with probtest, and this would required a more involved validation or investigation. Both were tested but only gave little gain out of the box so it was decided to not further invest time on this aspect. This may be investigated in a future work.

> With regards to the optimisation flags, a common optimisation strategy on NVIDIA hardware is to limit the number of registers in GPU using "-gpu=maxregcount:<n>". That helps to increase occupancy and can be applied on a per-sourcefile level to boost performance of poorly performing kernels. Has this been tested and found to not provide any performance improvement?

The maxregcount is set for the ecrad components and this helps with performance.

For the rest of the model, we did not find a single number that would produce better results, but hand-tununig every single kernel may improve the performance a bit, however it was consider that this was not worth the effort for this work.

Beyond the opt-rank-distribution described in Sec. 6.2.6, has the use of MPS to oversubscribe GPUs with multiple MPI ranks been explored? While this obviously reduces performance of kernels that are well saturating the hardware, it may help to improve occupancy for lower performing kernels.

A few tests were done with MPS using a previous version of the model and for a different slightly different configuration. This however did not bring much overall improvement and was not considered for this work. This may also be considered at a later stage.

I am unfamiliar with the Slurm setting "-distribution=plane=4" - could this be explained in Sec. 6.2.6?

The following sentence is added "This plane distribution allocates tasks in blocks of size 4 in a round-robin fashion across allocated nodes."

> The use of mixed-precision gives a surprisingly little performance improvement. IFS has achieved about 40% runtime improvement by switching to single precision (see https://doi.org/10.1002/qj.4181). This is likely limited by the number of fields that are still kept in double precision. What is the current share of operations/fields in single/double precision? Automatic tooling may help here to identify fields and operations that are sensitive to numerical accuracy, see for example https://github.com/aopp-pred/rpe or https://doi.org/10.1007/978-3-031-32041-5_20.

The authors also be believe that the few number of fields is one reason for the little improvement. We also note that for the COSMO model which was able to run in single precision, we also got close to 40% improvement on CPU but only about 20% on GPU. One possible explanation could be related to overhead of other operation such as integer arithmentic for example for index computation. There are several ongoing project in ICON to run entirely in single precision, however this raises challenges in terms of validation and are beyond the scope of this work.

The following sentence is added:"This performance improvement is somewhat less that one could expect, and may results from several aspects. First there are only a limited number of fields in ICON that are set to single precisions in the dycore. Another possible explanation is that the overhead of other operation such as integer arithmetic for example for index computation, might be taking a non significant part of the time, and become a limitation in mixed precision."

> The timing results reported in Table 1 are helpful in assessing performance gains from the described optimisations. I would recommend to add a bar plot that presents the runtime and improvements for each optimisation described in Sec. 6, thus illustrating the gains achieved. The presented numbers also include only dycore and physics: what runtime share is attributed to other components shown in Figure 1 (e.g., "Infrastructure" and "Diagnostics")?

A bar plot was added and the following sentence was added in 6 : For a typical NWP run, most of the run time is spent in the dynamics, about 50\% , and the physics, about 30\% which are reported in the table and figure. The rest is spend in the communication, 8\%, initialization, which includes the data transfers to the GPU, 4\% , diagnostics and output which are below 5\% each. The output is done asynchronously, and is to a large part overlapped with the computations.

Section 6.4 claims that this now enables "more efficient and cost-effective use of computational resources" - compared to what? It begs the question if the original objective when embarking on the GPU porting work has been achieved, e.g., is the production of forecasts now cheaper/faster/more energy-efficient than on CPU?

The sentence reference to the improvement due to opimmizatin as compared to the initial code without optimization. However on a more general view considering that the CPU and GPU sockets have similar power envelop one of the main gain of using GPUs is in terms of energy needed for a given configuration and size of the system. In fact some study, e.g. Cuming 2014, show that the gain in performance is even slighly larger when looking at the energy. The overall price is more difficult to assess since the MeteoSwiss system was acquired by CSCS togehter with a larger GPU system, but at the time of setting up the MeteoSwiss system this appeared to be the best solution also in terms of price. With the now much higher cost of some GPU hardware the factor 5.1x may not be enough for the initial acquisition only, however the argument of the energy will still hold. This is highlighed in the conclusion:

"This potentially allows for a much more compact and energy-efficient system than a CPU-only system."

The following typos/minor and remarks are corrected.

- l.70 "Further more" -> "Furthermore"

- Caption of Fig. 1: This should likely read "After Initialization on the CPU..."

- l. 311: "In contract" -> "In contrast"

- l. 507/508: two subsequent sentences start with "In particular"

**Reply to Anonymous Referee #2**

The authors would like to thank Referee 2 for the detailed reading and comments. A detailed answer to the comments can be found below.

> Section 1, lines 20-24: Having a GPU based model was mentioned as an advantage in the context of emerging Machine Learning for weather forecasting. The rest of the paper focuses on the computational performance gains from GPU porting, so I am wondering where the mention of ML application fits. Are there plans to incorporate ML algorithms and applications into ICON in the future, and how would this affect the porting strategy presented in the paper? The conclusion briefly mentions it, but it is not very specific.

The main gain is that having a GPU system for NWP allows MeteoSwiss the same GPU ressources to train ML models. There are no plans at MeteoSwiss to work on hybrid algorithms for the time beeing, but there are some initiative in the ICON community. The authors believe that the GPU port could be beneficial in an hybrid setup since both the physical based model and the inference could run on GPU avoiding the need of data transfer.

A sentence is added:

"Indeed in case of an hybrid setup both physical model and the ML inference could be run on CPU without requiring data transfer. The main synergy for MeteoSwiss currently is that the GPU infrastructure that was acquired for the NWP forecast can be used for training and inference of ML weather models."

> Section 2, lines 59-61: I find this sentence a bit unspecified for verification: "This approach, combined with advanced data structures and efficient communication protocols, ensures scalability and optimal performance on massively parallel super-computing architectures." What is the context and baseline for the claim that the data structures are advanced? Is it in comparison to what was applied in the same programming language but in earlier incarnations of the model, or the similar model? Are they advanced in comparison to other languages? As for the efficient communication protocols, some specifics on what is used and why would also aid understanding.

The sentence was reformulated to be more precise "The communication accross nodes is implemented using the Message Passing Interface (MPI) library and the model is scaling well up to thousand nodes~\cite{giorgetta2022}."

>Section 3.1, line 68: "directive-based approach", and "GPU-specific language"), I believe (compound adjectives).

This was corrected

>Section 3.1, line 69: "its" instead of "it's", I believe (possessive pronoun).

This was corrected

> Section 3.1 and later Section 4.3 (choice of porting approach and challenges): It is stated that "…approach was decided over a re-write in a GPU specific language like CUDA or a DSL, mainly because of it's (sic!) broader acceptance by the ICON community". Does this mean that the directives were added by hand by the ICON developers (domain scientists and RSEs, mentioned in Section 4.3)? I appreciate that re-writing an NWP model in something like CUDA is heavy-handed, however I am wondering what the main obstacle was against adopting a DSL approach, especially as it can facilitate porting by reducing the need for manual intervention. The section 4.3 mentions training domain scientists "in the basics of the GPU port, OpenACC, and tools for GPU verification", and that the ICON developer community "comprises many scientists without a formal computer science background". OpenACC and GPU port is not exactly the easiest thing to teach to such audience, and I am not sure that it would be more difficult to teach a DSL approach to them. I am not saying that the approach here was right or wrong, I am just wondering what motivated it (e.g. existing DSL tools not being mature enough to generate GPU parallelisations, or perhaps being difficult to teach, or there is uncertainty in maintenance / funding). I appreciate that when it comes to DSLs there is a decision whether to invest in the tool itself and who maintains it.

In fact there is an ongoing re-write project using a DSL, gt4py. A sentence was added in the introduction to mention this work. However beyond acceptance of a DSL approach there was also the aspect of readiness. On the MeteoSwiss side there was a critical timeline in view because of the life cycle of the HPC and modeling system which required to have ICON ready to run on GPU at a specific point in time. At the time we started to actively port the model OpenACC was the only demonstrated working approach, so this can be viewed as both a risk mitigation approach as well as a baseline for future implementation that may be more future proof.

> Section 3.1, lines 85-86: This can read as the conclusion only porting isolated kernels to GPU yields little benefit applies in atmospheric models in general, and I think this may not be entirely correct as the performance profiles or different models, and therefore optimisation strategies, can vary. I see that the paper from Adamidis et al. used ICON as the case study, so I assume the decision to go for the full port strategy here instead of porting individual kernels was heavily based on the performance profiling of that case? If so, it would be worth mentioning that.

The evaluation was indeed based on profiling the model, as well as measuring the ratio of transfering 3d fields typically used for every components as compare to the respective time of the kernels which is comparable to the time of an entire time step. This means that there is little speedup possible as long as even one components using several 3d fields at every time step is not ported to the GPU as the data transfer would be dominating the overall timing. This evaluation is prior to Adamidis et al. paper, and is also based on previous work done for porting the COSMO model. The authors believe this statement is generaly applicable to NWP and climate models, at least for architecture where GPU and CPU have separate memory. The following sentence is added:

In atmospheric models, the arithmetic intensity, i.e., the ratio of computation (floating-point operations) to memory accesses, is generally low~\citep{adamidis2025}. As a result, only porting isolated kernels to the GPU would incur a significant penalty of data transfer between the CPU and the GPU that cannot be amortized by the ported kernel execution time.

>Sections 3.1 and 3.2: If possible, I would advocate for placing Listing 1 to Section 3.2, as this is where the terms in Listing 1 are explained fully. The reference to Listing 1 in Section 3.1 could be adjusted as e.g. "see Section 3.2, Listing 1".

The listing 1 was simplified to only used basic openacc statement, and no ICON specific terms.

>tolerance validation with probtest.

This referee comment seems truncated.

>Section 3.2, lines 109-110: I am curious how the additional logical argument, lacc, is introduced into low-level shared routines. Would it be possible to add a small code listing to illustrate this?

A new listing is added showing a kernel with a conditional.

>Section 3.3: The introduction gives an overview of different optimization strategies for ICON, outlining when it is possible to apply them. I think some references to what strategies were utilised mostly in what parts of the model in the following sections (e.g. dynamics,

transport, physics) would be very useful. For instance, I would imagine that quite a few loops over horizontal sub-domains would have the same bounds (up to redundant computation for the difference).

The optimization described in section 3.3 have been applied in most part of the code, and are not specific to components. The sentence was modified as:

"Multiple general optimization strategies are considered for the initial GPU port of ICON and are applied through out the code when possible."

>Section 3.6.2, line 195: What is the "reduced grid" in the ecRad scheme?

The reduced grid is the coarse grid describe in the paragraph just before. The sentence was modified as follow:

"To reduce computational time and modestly decrease memory usage, radiation

calculations are performed on a coarser grid, so called reduced grid, approximately four times less

dense than the full-resolution grid."

>Section 3.6.2, Figure 2: It is not quite clear to me what the figure illustrates here. Are these two ICON arrays mapping into the same ecRad data structure? Or is it the same ICON array but mapping of its different nproma portions?

It is the maping of one icon array. The caption is reformulated as follow:

"Example of the mapping of an ICON array, of horizontal size nproma, to an ecRad array, of horizontal size nproma_sub, over two

    iterations of the radiation sub-block loop. This simplified sketch omits additional vertical

    levels, the wavelength dimension, and the many variables involved in

    the computation."

>Section 3.8, Figure 3 and lines 254-259: What are the red circles in Figure 3? It would also be useful to reference elements of the figure (red circles, blue and green squares) in the text referring to the figure, as it is not quite clear what happens when.

The following sentence was added to the caption of Figure 3: "The red dots represents different ensemble members of the Analysis."

In addition they are referenced in the text

>Section 3.8, lines 263-265: It would be useful to have some more specifics on the cost of data transfers between GPU and CPU, as it seems that it seemed to be acceptable in comparison to porting the Data Assimilation to GPU.

The measurement of the data transfer was added in the following sentence:

"The data transfer time for this configuration corresponds to 0.09\% of the total runtime."

> Section 3.9: Again, I would be curious about the mechanism and the cost of data transfers between GPU and CPU when outputting diagnostics at designated intervals. Would it be possible to provide some estimates and how they affect model times? Also, is the point of completing all the diagnostics calculations on GPU prior to sending the data to CPU a kind of "synchronisation" point for the model (or parts of it)?

We do not have a measure of the data transfer separated from the output, however the output is a little less than 5% of the runtime and includes the data transfer. The output being asynchronous a large part of the time is overlapped with computation. This information is added in section 6.

> Section 4.1, lines 289-290: "A short and computationally inexpensive configuration is used to run the perturbed ensemble and determine the expected spread of each variable". I assume the same configuration is calculated on CPU and GPU, as later indicated?

yes this is correct. The following is added "using the same configuration"

> Section 4.1, lines 293-295: "set of configurations which are supported on GPU using a reduced domain size. Every change to the model must pass this validation step to be accepted." Is the change to the model tested on the same reduced domain size? How is the reduced domain size chosen to be sure it is an adequate representative of behaviour on operational model sizes?

The model is running on a smaller domain, but the exact same configuration as the operational system. The sentence "For example one test is using the exact same configuration as the ICON-CH1-EPS ..." is added.

>Section 5: The two operational NWP ICON configurations presented here both refer to regional domains, as well as the results presented afterwards. Does this mean that the optimisation strategies presented in Section 3 were chosen for the regional configurations of the model? Or were they more general and applied for global and regional configurations? It would be good to clarify that.

The code path used for regional and global are for most of the code the same, although the MeteoSwiss setup was the main target for optimization, the optimizaiton have also been tested for some global setup, such as the operational global system of DWD.

A reference is added in the conclusion: "tested including one global configuration from Germany's National Meteorological Service, the Deutscher Wetterdienst (DWD)."

>Section 6.1: Is a single ensemble member run on two GPU nodes?

It can run on 2, but to have more margin and since not all optimization were ready when we started operation, one ICON-CH1-EPS is run on 3 Nodes. This was added in the description in section 5.1.

> Sections 6.2.1-6.2.9: Are the performance improvements for the total model run (including parts computed on CPU plus data transfers)?

yes. The sentence "The reported timings includes all data transfers..." was added in section 6.1

> Section 6.3: Are the scaling results presented here for the total model run (including CPU-computed parts and data transfer)?

Yes - the sentence "The reported timings includes all data transfers" in section 6.1 applys to all results. The text was adapted accordingly.